# Inequality and fairness with heterogeneous endowments

**Milena Tsvetkova** [1] *, **Oana Vuculescu**[2], **Petar Dinev**[3], **Jacob Sherson**[2,4], **Claudia Wagner**[3]

1 Department of Methodology, London School of Economics and Political Science, London, United Kingdom, 2 Department of Management, Aarhus University, Aarhus, Denmark, 3 Department of Computational Social Science, GESIS–Leibniz Institute for Social Sciences, Cologne, Germany, 4 Department of Physics and Astronomy, Aarhus University, Aarhus, Denmark

* m.tsvetkova@lse.ac.uk

## Abstract

People differ in intelligence, cognitive ability, personality traits, motivation, and similar valued and, to a large degree, inherited characteristics that determine success and achievements. When does individual heterogeneity lead to a fair distribution of rewards and outcomes? Here, we develop this question theoretically and then test it experimentally for a set of structural conditions in a specific interaction situation. We first catalogue the functional relationship between individual endowments and outcomes to distinguish between fairness concepts such as meritocracy, equality of opportunity, equality of outcomes, and Rawl's theory of justice. We then use an online experiment to study which of these fairness patterns emerge when differently endowed individuals can share their resources with others, depending on whether information about others' endowments and outcomes is available. We find that while visible outcomes lessen inequality by decreasing the statistical dispersion of outcomes across the group, endowments need to be visible for better equality of opportunity for the most disadvantaged.

**Data Availability Statement:** All data and analysis scripts are available from the figshare database at https://doi.org/10.6084/m9.figshare.21103021.v1.

**Funding:** This research was made possible through the generous support of the Volkswagen

## Introduction

People tend to recognize individual differences in physical strength, attractiveness, cognitive ability, intelligence, and motivation as a fact of life. Whether they consider them ascribed at birth or achieved over the life course, people value such characteristics and deem them relevant for individual achievements such as income, wealth, popularity, power, and social status. Yet, individual success and achievements are sometimes judged as unfair and unjust; for example, scholars, activists, and politicians often talk about inequality, implying a problem that needs to be addressed. When does heterogeneity in outcome-relevant characteristics result in fair outcomes?

The current study investigates this question both theoretically and empirically. We present a novel conceptual schema that explores the functional relationship between individuals' outcome-relevant characteristics, or endowments, and individuals' outcomes to synthesize prominent normative concepts of group-level fairness such as meritocracy, equality of opportunity,

Foundation (Grant Ref. 92 173). The game development was partially supported by the Carlsberg Foundation (Grant no. CF16–0593). The funders had no role in the study design, data collection and analysis, decision to publish, or preparation of the manuscript.

**Competing interests:** The authors have declared that no competing interests exist.

equality of outcomes, and Rawl's theory of justice. We then focus on two specific structural conditions–the visibility of individual endowments and the visibility of individual outcomes– and hypothesize how they affect individual behavior and interactions, and consequently, the expected pattern of fairness in social interaction groups. We test the predictions with an online experiment in which participants, randomly allocated to receive a different number of resources, repeatedly interact in groups by choosing whether and how to invest their endowed resources in others.

Quintessential to the social sciences, the problem of fairness spans the fields of political theory, sociology, and behavioral economics. Approaches range from normative theories of society-level solutions [1, 2] to empirical studies of individual-level subjective justice evaluations in allocation and negotiated exchange tasks [3–6], and equity considerations in cooperation games [7–10]. Our study bridges these different approaches and contributes to the existing literature in several ways.

First, regarding fairness, we shift the theoretical focus from subjective perceptions and normative argumentation to objective categorization. We propose a conceptual schema that synthesizes and contrasts different group-level patterns of fairness, given a distribution of outcome-relevant characteristics. The schema aligns with normative political theorists' approach to fairness as a macro-level phenomenon but also acknowledgs the plurality and context-dependence of fairness concepts, the cornerstone idea of work by empirical scholars of individual justice perceptions and evaluations. The proposed conceptualization can be used to guide research on inequality and fairness from different perspectives and methodological approaches–experimental tests in social interaction groups, surveys of individual allocation and redistribution preferences, as well as quantitative analyses of inequality in organizations and the general population.

Second, we consider group-level fairness not as a preference or a principle but as an emergent phenomenon. Like [11], we problematize the link between micro-principles and macro-patterns of fairness but we focus on bilateral exchange relations with free partner choice instead of unilateral allocation decisions. This allows us to approach interaction groups as complex social systems where individual preferences, behavior, and interactions do not simply aggregate to macro-outcomes but may have unintended consequences. Following the analytical sociology paradigm [12, 13], which emphasizes mechanism-based explanations for macro-level phenomena, we differentiate between the principles individuals employ to achieve fair interactions–e.g., charity, reciprocity, inequity aversion–and the group-level fairness patterns these unintentionally produce–e.g., equality of opportunity and meritocracy.

Third, we provide initial empirical evidence for how a specific set of structural conditions affects emergent group fairness. We conduct an experiment where individuals repeatedly choose to what extent and with whom to share their endowed resources, with the opportunity to establish reciprocal relations over time. The social interactions we model allow us to draw parallels to empirical research on reciprocal exchange from social psychology [14] and cooperation games from behavioral economics [15–17]. The experiment continues a growing line of research that employs cooperation games in fixed or endogenous networks to study the emergence and persistence of inequality in social groups [18–23]. We extend this research by distinguishing between endowments and outcomes, investigating the effects of the visibility of both, and presenting a more complex understanding of inequality.

Fourth, the experiment we develop and conduct advances social research methodology in that it uses gamification techniques to simplify actions and convey complex decision information to participants. In contrast to most previous experiments, which involve reading multiple pages of text-based instructions, processing information from previous rounds in the form of tables with numbers, and visualizing the interaction situation with circles and lines on a blank

background, our experimental game utilizes an interactive tutorial to simulate the game setting, animated visualizations to present decision-relevant information, and 3-D graphics and sounds to embed the decision situation in a compelling narrative. Although gamified experiments are time- and resource-intensive to develop, we believe they carry an immense potential to scale up experimental social science [24] by offering new ways to involve the general public, reduce the costs of monetary compensation, and increase participant engagement and retention.

Finally, the experiment generates new empirical knowledge about the effects of endowment heterogeneity. It confirms previous results about the effects on individual behavior but also yields novel findings about the group-level consequences. Most notably, we find that the visibility of endowments can provide equality of opportunity for the least endowed without erasing the advantage of the most endowed. The type of social interactions we study underlie primitive gift-giving economies but also knowledge sharing, advice giving, mentoring, money lending, and other forms of social exchange and mutual help. Our findings thus have relevance for schools, organizations, residential neighborhoods, and online communities, where individuals arrive with varying levels of knowledge, abilities, and experience and can accelerate their success and achievements by learning, collaborating, and cooperating with others. Inequality in experiences, performance, and outcomes is a significant concern in some of these settings. For example, two of the most fundamental missions of education institutions in contemporary Western societies include providing equality of opportunity and reducing achievement gaps [25]. Similarly, business organizations are becoming increasingly more concerned with reducing inequalities among employees in order to increase individual productivity, individual job satisfaction, and organizational performance [26, 27]. Our research gives tentative insights as to how sharing information about inherent disadvantages, such as a problematic family situation or learning disabilities, and outcome indicators, such as employee productivity statistics, student grades, or online users' skill levels and scores, could affect cooperation and the network of interactions and stir the group towards a desired outcome, depending on the notions of fairness the organization prioritizes and aspects of inequality it eschews.

## Fairness at the group level

Although the concepts "fairness" and "distributive justice" are often used interchangeably in the literature [10, 11], Rawls [1] delineates a useful distinction between the two: we talk about fairness in the context of cooperation, competition, and exchange relations between individuals without authority over each other and with the free choice whether to engage; in contrast, distributive justice pertains to unilateral authority towards choice-free individuals. This distinction is reflected in the fact that empirical research on distributive justice tends to address individual allocation preferences, perceptions, and judgements [3, 5, 6], while research on fairness tends to employ cooperation games in dyads and networks [7, 8]. Since our research focuses on bilateral interactions, we will use the term "fairness."

Despite this conceptual distinction, certain themes cut across research on both topics. One is the contrast between micro-principles and macro-conceptualizations [5, 11, 28]. Another is the acknowledgment (at least by the positivists) that multiple ideas of fairness and justice exist and that context matters [10, 29, 30]. Considerations such as individual needs, contributions, ability, and luck can be more or less prominent in different social contexts and situations. Here, we restrict our focus to one specific individual factor: an outcome-relevant endowment. This can be any exogenously determined resource or ability due to advantageous upbringing, genetic inheritance, or pure luck that determines the individual's achievements and rewards. We purposefully do not specify whether the endowment corresponds to an ascribed or

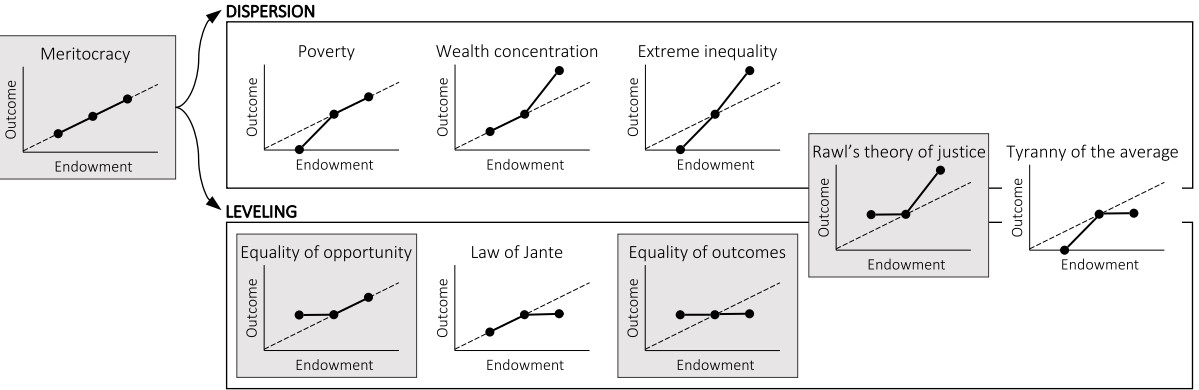

**Fig 1. Schematic representation of macro concepts of fairness (shaded in gray) based on the relation between individual endowments and outcomes.**

achieved characteristic because many characteristics, such as education and intelligence, can be considered both, or can be perceived as one or the other, depending on the context. This allows us to compare and contrast a wide range of macro-concepts of fairness simply on the basis of the relationship between endowments and outcomes, without the need to specify the context or make normative statements.

In Fig 1, we propose a schema that outlines the main concepts of fairness and inequality through the functional relationship between individual endowments and outcomes. By definition, this conceptual schema is reductive and cannot capture the depth and nuances of centuries of philosophical thought and social theory on these topics. Nevertheless, it is useful for comparing and juxtaposing the different concepts. Our approach parallels Jasso's [11] discussion of the just reward function $y = h(x)$, where $y$ is the just reward for her and "outcome" for us, and $x$– the reward-relevant characteristic, or "endowment" in our case. However, while Jasso assumes that the precise functional form $h(x)$ is a question of individual belief and judgement, we assume it is a signature of a well-established theoretical concept of fairness or inequality.

The schema is anchored on the simplest pattern where the relationship between endowments and outcomes is linear. We will refer to this pattern as "meritocracy." Meritocracy is grounded in the principle of equity, conveyed in the idea of "giving every person their due" [5]. Typically, the endowments that merit proportionate rewards are assumed to be achieved characteristics such as intelligence, talent, motivation, educational qualifications, and effort, as opposed to inherited wealth, social class, or social categories such as gender and race. Nevertheless, as we already argued above, many of the characteristics we commonly perceive as achieved can be traced back to privileged socio-demographics and parentage. In fact, the most prominent lines of critique of the principle of meritocracy rest on this argument [31, 32]. Here, we will use the term abstractly, as a shorthand description for a linear relation between endowments and outcomes. The ideal size of this relationship (the slope of the dashed line in Fig 1) will depend on the specific context, subjective preferences, and/or normative deliberations. For our argument here, we will simply assume some arbitrary positive slope.

We suggest that deviations from the ideal proportionality of what we here label "meritocracy" can go in two directions: **dispersion** and **leveling**

In the first case, outcomes are significantly more exaggerated than the underlying heterogeneity (see the panels labeled poverty, wealth concentration, and extreme inequality in Fig 1). These situations are typically considered problematic, as evidenced by the recent public outrage at the extremely high CEO-to-worker pay ratio [33], as well as the increasing attention to

rising inequality at country level [34]. In the second case, the outcomes can level off, diminishing some of the distinctions between differently endowed individuals. We note that leveling is not the inverse of dispersion but a separate phenomenon; it is possible to have outcomes that are dispersed but also diminish distinctions by endowment, as well as compressed distributions of outcomes that respect ordering by endowment.

Apart from meritocracy, most concepts of fairness rely on some degree of leveling. The idea of equality of opportunity posits that leveling is desirable when it means that the most disadvantaged can improve their lot. Equality of opportunity implies that individuals should not be limited by characteristics ascribed at birth to advance in society and achieve higher economic and social rewards [35]. Although not a widely accepted idea of fairness, another way to achieve leveling is to decrease the outcomes of the most endowed. This is captured by the Scandinavian norm known as Law of Jante and the related idea of the tall poppy syndrome–the idea that standing out by outdoing everyone else is undesirable. Combining equality of opportunity with the Law of Jante, equality of outcomes represents another main concept of fairness, albeit controversial and highly politicized [36–38].

Finally, John Rawls' theory of distributive justice stands out because it allows for dispersion of outcomes for the most endowed as long as it is compensated by leveling for the least endowed. Specifically, Rawls' Difference Principle allows for inequalities as long as the least endowed are guaranteed fair equality of opportunity that compensates for their naturally occurring disadvantage [1, 2].

In sum, the conceptual schema allows us to identify a pattern of inequality or fairness by measuring the dispersion and leveling of outcomes at different endowment levels. The schema is general enough to be useful for a wide range of empirical research, including studies of individual perceptions of inequality and fairness in exchange situations or judgements in allocation tasks. Here, we will apply it to categorize the group-level patterns that emerge from social interactions. We model a specific social interaction situation where endowments are represented as a fixed number of resources people have access to that they can invest in themselves or in others. For instance, in school and organizational settings, students and employees may have different productivity, knowledge, or skill levels and they can choose to use these to advance their own outcome or help others. As a natural start, we focus on the effect of two simple structural conditions: we study how information about others' endowments and outcomes affects individual partner choice and resource sharing decisions, and consequently, the fairness of the group outcomes. We investigate four situations: there is no information about others (NI), others' endowments are visible (E), others' outcomes are visible (O), and both endowments and outcomes are visible (EO). In the interaction situation we model, individual outcomes increase with every received resource but the accumulated outcomes cannot be reinvested. This precludes increasing-return processes, which cause high dispersion of outcomes and extreme inequality [39]. Hence, our expectations do not concern dispersion, but we revisit this point in the discussion.

## The effects of endowment heterogeneity on group-level fairness

Humans are hardwired to pursue fairness in their immediate interactions [40–42]. Individual preferences for fairness, however, may not necessarily result in fair outcomes at the group or societal level [43–45]. Even if they do, it is not obvious which conditions make either of meritocracy, equality of opportunity, equality of outcomes, or Rawlsian fairness more likely than the other. We posit that different information about others constrains how individuals choose exchange partners and accomplish fair dyadic interactions, which in turn determines what pattern of fairness the group achieves.

In the context of gift giving and helping that we consider, two differently endowed individuals can achieve and maintain fair exchange in three different ways: via charity, reciprocity, or inequity aversion. With charity, an individual would help someone who is disadvantaged or unfortunate regardless of the individual's own state and without expecting anything in return; an individual could give charitably due to pure altruism, sympathy, a "warm glow" feeling, social pressure, or guilt [46, 47]. Following reciprocity, an individual would return someone's action or gift in kind, regardless of who is more advantaged or fortunate [48–50]. And under inequity aversion, an individual would attempt to compensate for existing differences by choosing to act in a way that equalizes the outcomes [51, 52]. For example, the disadvantaged individual would give less than they could to the advantaged individual, while the advantaged individual could give back more than they receive.

In the NI condition, when endowments and outcomes are unknown and others appear identical, individuals will select their exchange partners randomly and insist on reciprocity, matching others' investments in absolute terms and expecting the same in return. For example, prior research shows that when heterogeneity is unknown and the interaction network fixed, the worse endowed give the same absolute amount (and more as a percentage of their endowment) as the better endowed [16, 17, 53]. If individuals can freely choose whom to interact with, however, the better endowed can simply find more partners to reciprocate with, which will result in equal cooperation levels among the different subgroups. Consequently, the individual outcomes will correspond to the underlying endowments. We thus hypothesize: **(H-NI)** **When no information about others is available, the group achieves meritocracy**

We remind the reader that we use the term meritocracy broadly to denote a linear relationship between outcome-relevant characteristics and outcomes; nevertheless, we recognize that the term will not apply normatively in the case of some endowments, such as inherited wealth.

In condition E, which is when endowments are visible, the effects on behavior are less certain. On the one hand, individuals might seek the better endowed, expecting that the better endowed will be inequity averse and contribute more than they receive. Previous studies have revealed that the better endowed can indeed be more cooperative and generous under some circumstances [54–56]. However, other research suggests that while the worse endowed act in accordance with inequity aversion, the better endowed insist on reciprocity and contribute not in proportion to their endowment but equally in absolute terms [15, 57]. Moreover, individuals are willing to reciprocate gifts from the better endowed, even if they are not entirely equitable, because they confirm their expectations [58]. In sum, prior work suggests that even if the better endowed are charitable or inequity averse towards the less endowed, they are likely to preserve their advantage due to attracting more exchange partners. Considering the combination of charitable giving or inequity aversion towards the less advantaged with preferential attachment towards the more advantaged, our hypothesis is: **(H-E) When others' endowments are visible, the group achieves equality of opportunity or Rawlsian fairness.**

In condition O, when only outcomes are visible, the result at the group level will depend on whether the wealthy behave in ways that reinforce their advantage or whether they compensate for their advantage by giving charitably to the poorer. Previous research suggests that the wealthy are less generous [59, 60] and specifically, less cooperative when wealth is visible [19]. Prior work has argued that an acquired sense of deservingness makes the rich individuals more selfish while limited experience makes the poor individuals tolerate selfishness [61]. Such blatant exploitation by the wealthy, however, is more likely to occur when individuals cannot freely choose their interaction partners or cannot reciprocate directly. In the exchange situations we investigate, we suspect inequity aversion will be more likely to prevail, such that the poor avoid giving to those who are wealthier, while the wealthy, driven by generosity or

guilt, give more to the poor. As a result, outcomes in the group will equalize. We thus expect:
**(H-O) When others' outcomes are visible, the group achieves equality of outcomes.**

Finally, in condition EO, we expect effects from information about both endowments and outcomes, as described above. The resulting group pattern will depend on whether either dominates or whether and how the two interact. On the one hand, individuals generally accept impartial procedures such as lotteries as fair [44, 62] and tend to avoid rank reversal [63]. Thus, knowledge of the endowments, when the outcomes are also visible, could result in outcomes that are linearly proportional to the endowments. On the other hand, inequity aversion by wealth could dominate the effects from knowing the endowments, which will result in highly equalized outcomes. Our final prediction is thus: **(H-EO) When both endowments and outcomes are visible, the group achieves equality of outcomes, equality of opportunity, or meritocracy.**

## Related work

To test the predictions empirically, we use an experiment in which we randomly assign participants to different resource levels and let them interact repeatedly in groups. We manipulate the visibility of endowments and outcomes and investigate how outcomes are distributed at the end of the interaction.

Similarly to [19], we study the effect of information about others on inequality but we expand on their work in several crucial ways. First, the authors model interactions with the N-person Prisoner Dilemma's game, which precludes direct reciprocity. In contrast, we model interactions as reciprocal exchange, in the form of directed, person-specific gift giving [14, 20, 21, 64, 65]. Second, [19] introduce heterogeneity via initial wealth and then manipulate the visibility of wealth. In contrast, our experimental task was designed to explicitly distinguish between endowments, or the number of resources available per round, and outcomes, or wealth. This allows us to conceptually differentiate between heterogeneity and inequality.

Like us, [20] study reciprocal exchange and differentiate between endowments and outcomes. However, they assume that neither endowments nor outcomes are visible, and that the networks are fixed and do not change. The authors manipulate the assortativity by endowment in the network and find that although reciprocity does not differ, outcomes are more dispersed when those with higher endowments are connected to each other. The study, however, assumes that a higher endowment is correlated with a higher number of exchange partners and this may not necessarily occur in a dynamic network where individuals choose how much and with whom to cooperate. In our study, we don't fix the structure of the networks, letting them emerge endogenously instead. Further, we analyze inequality both in terms of how outcomes are dispersed and how they correspond to the underlying endowments, which gives us a more comprehensive and nuanced view of inequality and fairness.

## Materials and methods

The study was approved by the Aarhus University Research Ethics Committee, CR no. 31119103. All participants were adults and provided written informed consent to participate in the study. The anonymized data and the Python and R scripts used for the analyses are openly available [66].

### Experiment

In the experiment, participants play a game equivalent to reciprocal exchange [14, 20] but with endogenous partner choice. Participants play in groups of 16–20 and over 20 rounds (the exact number of interaction rounds is unknown to them). Every round, each player receives a fixed

number of resources and decides how many of these resources to invest in themselves or other players. To simulate endowment heterogeneity, at the beginning of the game, we randomly assign about one third of the players to receive 2 resources every round, about one third to receive 4, and the remaining third to receive 6. Players obtain 10 points for each resource they invest in themselves and 15 points for each resource they receive from someone else; uninvested resources do not yield points. Thus, investment in others is collectively beneficial but individually risky since it may not be reciprocated. Players can choose to invest in any other player in the group and to keep track of their interactions, they can see who gave to them and to whom they gave in the previous round. Players do not, however, have information about interactions they were not involved in. In treatment E, players can see the number of resources per round others have been allocated, in treatment O, they can see the cumulative score of the other players, in treatment EO they see both, and in NI–neither. In all treatments, players can see their own resources and score. They are also informed that everyone starts with a score of 0 but some players receive two resources, some receive four, and others receive six; players do not know the exact resource distribution, however, except when they observe it in the E and EO treatments.

We relied on gamification techniques to make the game more engaging and compelling. We designed the game with a visually stimulating setting, simple and intuitive player actions, graphically presented information, and an interactive tutorial (S1 Fig). The premise of the game, which we called Urbanizer, is that players are property developers who compete to build up and develop their city block with their own resources or resources they receive from others. We developed Urbanizer in collaboration with Science At Home, a citizen science initiative based at Aarhus University. Additional details about the game instructions and mechanics are available in S1 Text.

We ran the experiment with participants recruited from the online crowdsourcing platform Amazon Mechanical Turk (AMT) in the period October 2019 –February 2020. On each day with experimental runs, we first recruit a large pool of participants with a short $0.50 task in which users read information about the study, give consent, and complete a four-question demographic survey. This task informs them when a session is scheduled during that day and that they need to wait for an e-mail with further instructions. About 15–30 minutes before the scheduled time, we use the AMT API to contact the recruits and send them the link to the game with additional instructions.

First, participants are asked to complete an interactive tutorial that leads them through the game's simple rules. Then, they join a virtual waiting room and as soon as 20 players show up, a new game starts. When at least 16 but fewer than 20 participants show up, participants have to wait 5 more minutes in case additional others decide to join. The game lasts 20 rounds (participants do not know the exact number) and each round after the first is 30 seconds long. The first round lasts 45 seconds because it starts with a pop-up window that describes that endowments are not equally distributed but all players start from a score of zero. As per the instructions in the invitation e-mail, when the game ends, participants go back to AMT to answer a brief survey about their understanding of the game and claim payment. Participants are awarded $2 for completing the game and a bonus based on their performance in the game equal to $(final score / 300). The minimum earnings were $3.38, the maximum–$8.88, and the mean–$5.16. The tutorial and the game typically took less than 20 minutes to complete, except for participants who had to wait longer for a group to form.

We have observations of 777 participants in 40 groups, with ten groups in each of the four treatments. The sample is restricted to residents of the USA and Canada and consists of 55.5% males with a mean age of 36.2. The modal education is Bachelor's degree and the modal annual household income is in the category $40,000–70,000.

We originally ran 43 groups but exclude three groups from the analyses due to technical problems that led to repeated participants, severe time delays, or the game crashing. About half of the groups we analyze (22 out of 40) have at least one and up to three players drop out before the game finished. Many of these players contacted us to say that their game was interrupted due to connection problems and thus, we assume that these dropouts are random and not related to the treatment, the player's assigned endowment, or their score. We use a discrete-time survival analysis that predicts dropping out with treatment, endowment, and score standardized per round to confirm that our assumption is justified (S1 Table). The only factor to approach statistical significance is the player's score, but this effect could also be explained with the player getting distracted and not actually playing the game, which would both lower their score and make them more likely to leave. In addition to the dropouts, two participants who completed the game had limited capabilities due to software incompatibility. Apart from the temporal exponential random graph models, the analyses we report here exclude the dropouts and the two problematic players. However, the group-level findings replicate even if we include them.

## Testing for dispersion and leveling

To test the hypotheses, we operationalize the concepts of dispersion and leveling separately and compare them between treatments. For dispersion, we evaluate the statistical dispersion of outcomes against the baseline statistical dispersion expected from the endowments. For leveling, we quantify the distinction in outcomes between different endowment levels.

We measure statistical dispersion with the Gini coefficient for the distribution of scores. The Gini coefficient is defined as half of the relative mean absolute difference:

$G = \frac{\sum_{i=1}^{n} \sum_{j=1}^{n} |x_i - x_j|}{2n^2 \bar{x}}$, where $x_i$ is the score of individual $i$, $\bar{x}$ is the average score, and $n$ is the number of individuals in the group. A Gini coefficient of 0 indicates perfect equality where all individuals have equal scores, while 1 indicates perfect inequality where only one individual has a non-zero score. The Gini coefficient is the most common measure of inequality but nevertheless has some limitations. In particular, it is sensitive to changes among those in the middle of the distribution and not so much to those at the lower end of the distribution [67]. We note that the Gini, or any other measure of statistical dispersion, does not perfectly map on our concept of "dispersion" as an indication of inequality. Within our theoretical framework, a higher Gini for the outcomes does not indicate inequality unless it is higher than the Gini for the endowments. Nevertheless, lack of inequality does not imply fairness. A lower Gini indicates less variability for the outcomes but whether this variability is fair will depend on how the outcomes correspond to the underlying endowments.

We measure leveling, or the reduction of outcome distinctions by endowment, by using Cliff's delta to compare the scores of individuals belonging to two different endowment levels. Cliff's delta is a non-parametric measure that quantifies the difference between two groups of observations [68]. In practice, it estimates how often the values in one group are larger than the values in another group: $d = \frac{\sum_{i,j} [x_i > x_j] - [x_i < x_j]}{mn}$, where the first group of size $m$ contains values $x_i$, the second group of size $n$ contains values $x_j$, and the Iverson bracket $[P] = 1$ if the statement $P$ is true and $[P] = 0$ otherwise. The measure is linearly related to the more common Mann-Whitney $U$ statistic but has the advantage that it provides the direction of the difference and an intuitive scale between 1, 0 and –1 that does not depend on the number of observations. In the current context, a Cliff's delta of 0 indicates that outcomes do not differ for individuals with either endowments, while 1 indicates that the better endowed individuals in the group always obtain higher outcomes than the worse endowed individuals.

To statistically test the differences between treatments, we analyze the group outcomes in terms of the individual scores at the end of the game. Ignoring the temporal persistence of outcomes reduces the power of the analyses but significantly simplifies our modeling strategy, yielding more intuitive, interpretable, and robust results. We calculate both the Gini coefficient and Cliff's delta from the final scores in each of the 40 interaction groups in the experiment and then compare treatments with the Mann-Whitney $U$ test. This is a non-parametric test of the null hypothesis that a randomly selected value from one treatment is equally likely to be less than and greater than a randomly selected value from another treatment. With four treatments, we conduct six pairwise comparisons. The ten groups per treatment give us 80% power to detect a Gini coefficient difference of 0.1 (assuming normal distribution with standard deviation of 0.05) and a Cliff's delta difference of 0.2 (assuming somewhat larger standard deviation of 0.1). Since our tests are hypothesis-driven and the six comparisons are non-independent, we refrain from correcting for multiple testing [69].

## Individual-level analyses

To acquire a better understanding of the behavior that drives the observed group-level patterns, we conduct additional analyses at the individual level. The interactions in each group can be represented as a dynamic directed network that evolves over 20 discrete rounds. Each link in the network indicates that giver $i$ gave one or more resources to recipient $j$ during a particular game round. The links are not independent because they are influenced by the grid structure in the game, the links in the previous round, and the players' endowments and scores. The scores are particularly problematic since they are endogenous to the network structure and dynamics. To account for these interdependencies, we use temporal exponential random graph models (TERGMs). The exponential random graph model (ERGM) is a statistical model that predicts the likelihood of a link in a network given a set of node-level, dyad-level, and structural terms and the TERGM extends the ERGM to longitudinal network data [70, 71]. A more technical description is presented in S2 Text.

We fit a treatment-specific TERGM model for each group and then conduct a meta-analysis over the ten groups in each treatment to estimate the common effect size in our experiment. The models we fit control for the tendency to give, the stability of edges between consecutive rounds, and triadic closure due to possible bias towards immediate neighbors on the grid. They also account for the fact that better endowed individuals can give more and that "wealthy" individuals can exhibit different giving patterns. We operationalize charitable giving by including a term for the recipient's endowment in the E and EO models and a term for the recipient's score in the O and EO models, where scores are standardized by round for each group. The coefficient for this term will indicate the extent to which a participant is more likely to give to someone with a low endowment or score, controlling for everything else. We operationalize reciprocity as the tendency to return a gift received in the previous round. In TERGM, this is the term "delrecip", which is implemented by including the transposed adjacency matrix from the previous round as a dyadic covariate. Finally, we operationalize inequity aversion as the tendency to reciprocate a gift from the previous round depending on how the giver's endowment and score compare to the focal node. We model this effect by adding as a dyadic term a matrix that is similar to the one used for "delrecip" but which has entries of 1 if $j$, who gave to $i$ in the previous round, has higher/lower endowment/score than $i$, and 0 otherwise. This effect will tell us the extent to which reciprocity is different in these cases compared to the reciprocity estimated when $i$ and $j$ have the same endowment and/or similar score. As before, we only include these effects when alter's endowment or score is visible in the treatment.

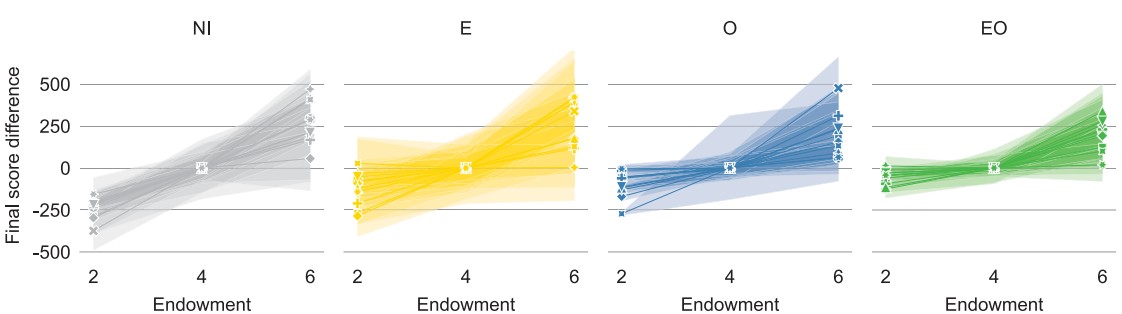

**Fig 2. The relation between individual endowments and final scores in the experiment suggests that outcomes are less dispersed when they are visible (O and EO) and that the distinctions between the least endowed and the rest are most palpable when no information about others is available (NI).** The figure depicts mean final score with 95% confidence intervals for each endowment level per group (different marker symbols correspond to different groups). Scores are centered on the mean score of the players with Endowment = 4.

We input the coefficients and standard errors estimated in the TERGMs for each group into fixed-effect models in order to estimate the true underlying effect of each treatment [72]. The fixed-effect model is appropriate for our case because the different groups within a treatment are functionally identical and we aim to compute the common effect size, rather than generalize beyond our specific experimental setup [73].

## Results

Fig 2 follows the format of Fig 1 to plot the relation between individual endowments and outcomes using data from the final round of the experiment (more detailed results for each group are shown on S2 Fig). From this simple depiction, we can see that when outcomes are visible, they end up seeming more equalized and less variable. We also observe that the distinction in outcomes between the least endowed and the average endowed is most prominent in the no-information condition. However, what is missing is the baseline slope of the proportional relationship against which we can evaluate the direction of the deviations. To quantify the observations from Fig 2 and test the hypotheses statistically, we assess the fairness of the group-level outcomes with respect to dispersion and leveling separately. Specifically, we evaluate 1) the statistical dispersion of individual scores and 2) the distinctions in scores by endowment.

### Dispersion

We measure the statistical dispersion of outcomes with the Gini coefficient for the distribution of scores. We also estimate the baseline statistical dispersion expected from the underlying heterogeneity by calculating the Gini coefficient for the distribution of endowments. Since the Gini coefficient is invariant to uniform scaling, this is equivalent to the scenario where no one cooperates and invests all resources in themselves, as well as the scenario where everyone cooperates at 100% and reciprocation is perfect. With a typical setup of six players of endowment two, eight players of endowment four, and six players of endowment six, the baseline Gini is 0.21.

We find that, in the experiment, the Gini coefficient is overall quite low, in the range 0.04–0.19 (Fig 3). Importantly, it is always lower than the Gini coefficient for endowments. As we explain in the introduction, the reason for this is that the interaction situation we model—reciprocal exchange with free partner selection—precludes increasing-return processes and exploitation but enables direct reciprocity and investment in long-term relationships; as a result, the outcomes tend to exhibit a more central tendency than the distribution of endowments.

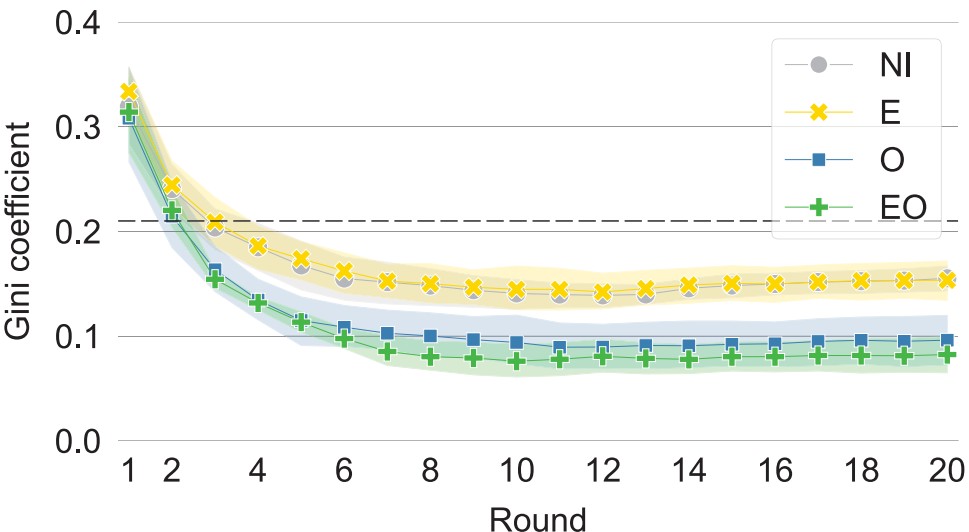

**Fig 3. The statistical dispersion of outcomes is lower when outcomes are visible (O and EO).** The figure shows means and 95% confidence intervals for the Gini coefficients of players' scores in the treatments with no information (NI), visible endowments (E), visible outcomes (O), and visible endowments and outcomes (EO). The dashed line represents the Gini coefficient expected from the distribution of endowments.

Overall, the small Gini values suggest that none of the treatments produce inequality in terms of poverty and wealth concentration, as we define it in Fig 1. Still, we find that the Gini is lower in the treatments with visible outcomes than the treatments without and this difference is statistically significant in the final round (Mann-Whitney $U = 12$, $p = 0.002$ for O–NI; $U = 14$, $p = 0.004$ for O–E; $U = 3$, $p < 0.001$ for EO–NI; $U = 0$, $p < 0.001$ for EO–E). This supports H-O, which predicted that outcomes will be more equalized when they are visible.

## Leveling

Two groups with similar statistical dispersion of outcomes may still differ in terms of leveling. Leveling occurs when the distinctions between differently endowed individuals are diminished. To identify leveling, we investigate the extent to which participants with higher endowments have higher scores (Fig 4). We estimate the correspondence between endowments and outcomes using Cliff's delta, an effect size statistic that measures how often the values in one distribution are larger than the values in another distribution. Confirming our observation from Fig 2, we find the highest correspondence between endowments and outcomes in the no-information treatment. Here, the better endowed individuals achieve consistently higher outcomes, which leads to a Cliff's delta that is closest to 1. This indicates that, in the absence of information, a meritocracy-like system emerges in which endowments and outcomes are linearly related, in line with H-NI.

We also observe that the difference in the final score distributions of the least endowed (endowment = 2) and the average endowed (endowment = 4) is smallest when endowments are visible (for final scores, Mann-Whitney $U = 6$, $p < 0.001$ for E–NI; $U = 26.5$, $p = 0.041$ for E–O; $U = 9$, $p = 0.001$ for EO–NI; $U = 29$, $p = 0.006$ for EO–O). These are also the treatments in which the least endowed give away a significantly higher proportion of their endowment than the others (S3 Fig). Thus, the treatments with visible endowments result in the highest equality of opportunity because in them, the least endowed can overcome their disadvantage and achieve higher rewards. This result was among the predicted outcomes in H-E and H-EO.

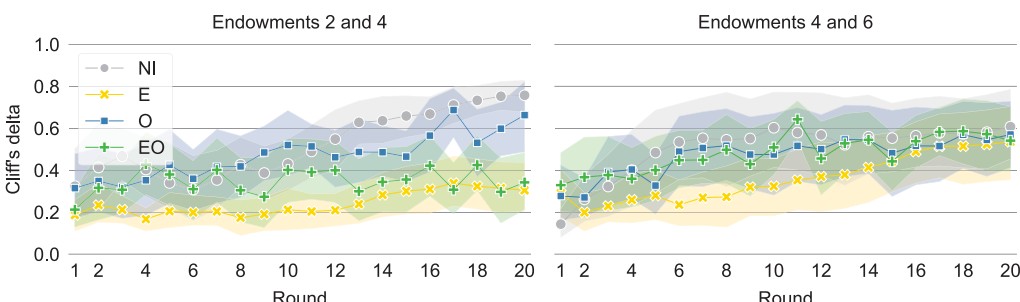

**Fig 4. Differences between the outcomes of the least endowed (Endowment = 2) and the average endowed (Endowment = 4) are lowest when endowments are visible (E and EO).** The figure shows means and 95% confidence intervals for the Cliff's delta statistic for scores between players with endowment two and four (left) and four and six (right) in the treatments with no information (NI), visible endowments (E), visible outcomes (O), and visible endowments and outcomes (EO). The maximum value of 1 occurs when all the players with the higher endowment have higher final scores than all the players with the lower endowment; 0 occurs when the players with the higher endowment are equally likely to have higher as lower final scores than the players with the lower endowment.

## Individual behavior

Finally, we investigate the extent to which individual behavior consistent with charity, reciprocity, and inequity aversion drives the observed differences at the group level. To provide statistical evidence, we present results from models that account for the network and time interdependencies in the interaction groups in our experiment. The models predict the likelihood, shown as log-odds in Fig 5, that $i$ gives at least one resource to $j$. The reference category in the models is a player with endowment of four resources and score within 0.5 standard deviations from the mean score in the group for the round.

We hypothesized a linearly proportional relation between endowments and outcomes when no information about others is available (H-NI) because individuals will focus primarily on reciprocity then. Consistent with this assumption, we find that the tendency to reciprocate is highest in the NI treatment (effect "$i$ reciprocates last round's gift" in Fig 5). This result is visible also when we simply track the proportion of edges that are mutual (S4 Fig).

We hypothesized two possible outcomes when endowments are visible (H-E) because we suggested a tension between inequity aversion and preference for the better endowed. We find some indication that the least endowed choose the most endowed in the first round (S5 Fig) but this preference does not persist unconditionally (the effect "$j$'s endowment = 6" in Fig 5 is not significantly positive). Instead, controlling for the fact that the better endowed can give more (the effects "$i$'s endowment"), the models reveal that participants are more likely to reciprocate to those with higher endowment and less likely to reciprocate to those with lower endowment than their own. The difference is particularly pronounced in the EO treatment. This result suggests that participants seek to establish more durable relations with the better endowed. At the same time, however, we observe giving charitably to the least endowed (see "$j$'s endowment = 2" in Fig 5). These contradicting behaviors explain why visible endowments erase the disadvantage of the least endowed, without negatively impacting the outcomes of the most endowed.

We assumed that inequity aversion will be prominent when outcomes are visible, resulting in greater equality of outcomes (H-O). Indeed, we find evidence that participants give remarkably more to the poor (S6 Fig). This is because participants are both more likely to give to someone whose score is too low (effect "$j$'s score $< -0.5$std" in Fig 5) and more likely to reciprocate a gift from someone who is poorer ("$i$ reciprocates if $j$'s score $< (i$'s score$-1$std)"). In other words, we find evidence for both charity and inequity aversion towards the poorer, which explains the less dispersed outcomes in the O and EO treatments.

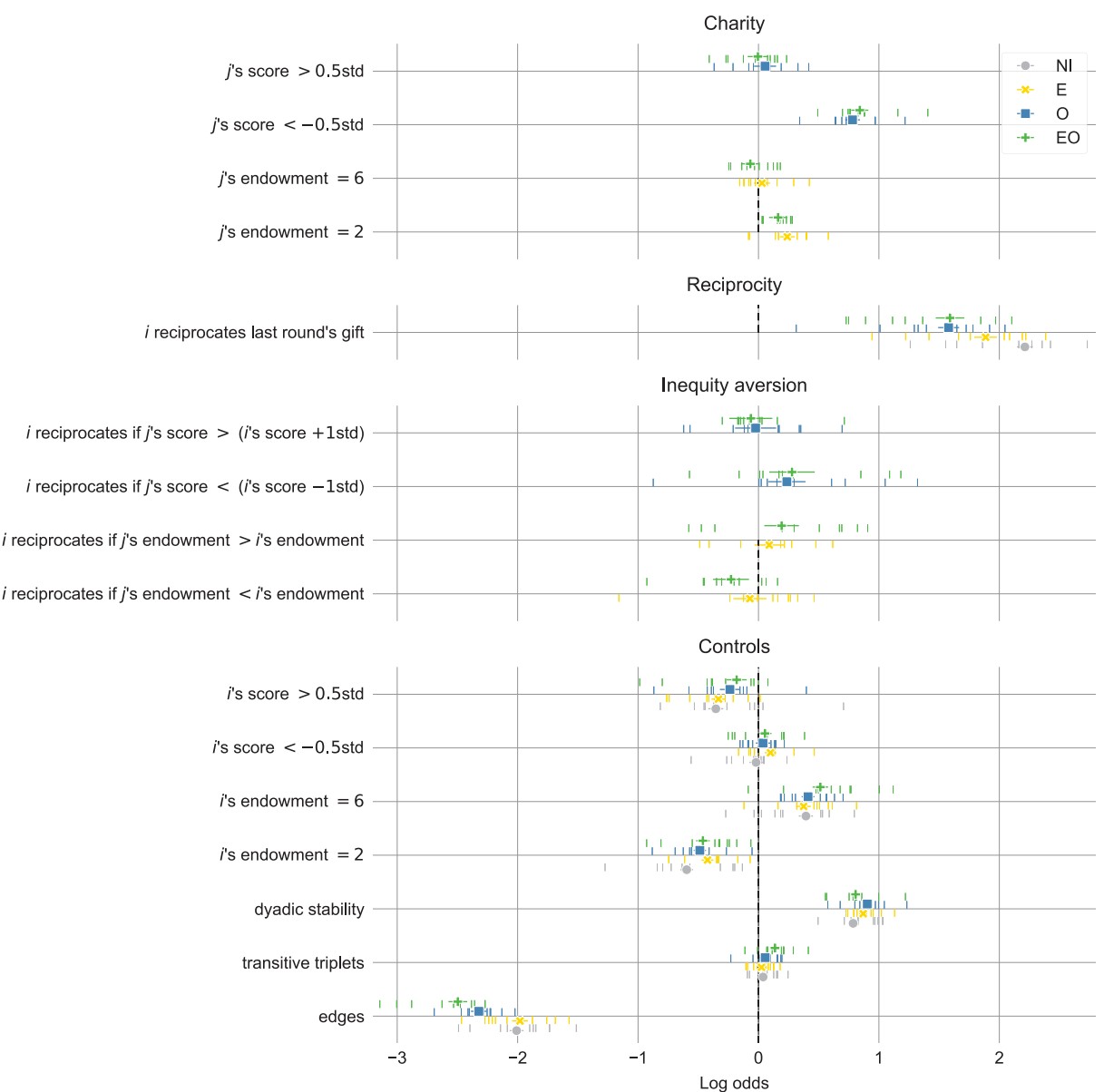

**Fig 5. We find statistical evidence for higher reciprocity in the absence of information about others (NI), charity towards the poor when outcomes are visible (O and EO), and charity towards the least endowed when endowments are visible (E and OE).** The results also show inequity aversion towards the poorer when outcomes are visible (O and EO) but higher reciprocity towards the better endowed when endowments are visible (E and EO). The figure shows effect estimates and 95% confidence intervals from meta-analyses with fixed-effect regression models of log-odds coefficients and standard errors estimated in temporal exponential random graph models. The vertical bars show group-specific estimates.

Finally, as we assumed in relation to H-EO, the behavioral effects from visible endowments and visible outcomes combine in the EO treatment. Thus, charitable giving towards the least endowed combines with charitable giving and inequity aversion towards the poorer to overcome the temptation to secure better endowed partners and produce outcomes that are more equalized overall and particularly advantageous to the least endowed.

For additional evidence on the motivations guiding individuals' decisions, we analyze participants' answers to an open-ended question asking "What was your strategy in the game?".

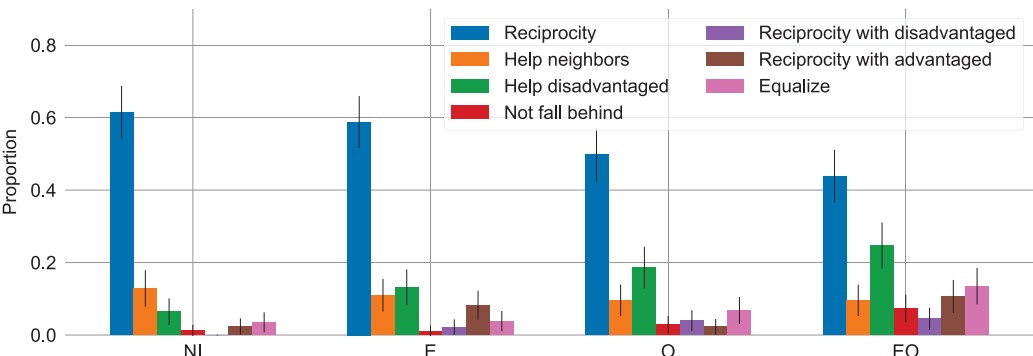

**Fig 6. Participants report using simple reciprocity most often when no information is available (NI).** When endowments are visible (E and EO), reciprocity specifically with the advantaged becomes a prominent strategy too. Helping the disadvantaged and giving in order to equalize outcomes but also to prevent oneself from falling behind are most prominent when both endowments and outcomes are visible (EO). The figure shows the proportion of participants in each treatment who mention one of the listed strategies in their free-text-entry response to the question "What was your strategy in the game?" The error bars show 95% confidence intervals.

This question was included in a short survey participants had to answer immediately after playing the game and before claiming their payment. One of the authors manually coded participants' answers (N = 708), identifying seven prominent strategies as shown in Fig 6; another author independently coded the data again for the same seven strategies. The intercoder reliability was satisfactory as estimated by Krippendorff's alpha for the edit distance between the 0–1 strategy vectors at 0.833; conflicting codings were resolved by mutual agreement between the two coders.

Comparing the differences in the proportion of participants in each treatment who mention a specific strategy confirms the main results from the statistical models above but also reveals new insights. Specifically, we find that reciprocity was most common as a strategy in the no-information treatment (NI) and that establishing reciprocal relations with advantaged partners in particular becomes significant when endowments are visible (E). For instance, one participant reports "I tried to share my resources with blocks that had access to more resources than me, in hopes that they would return the favor" and another: "I went for ones with highest resources to begin with then to keep loyalty, to those with a trade route." When outcomes are visible (O and EO), the intention to help the disadvantaged and the intention to equalize everyone's outcomes become particularly salient: e.g., "I wanted to prop up the lowest scoring blocks so that they could get more points" and "Spread the wealth!" Interestingly, when both endowments and outcomes are visible, we also find a non-zero proportion of participants who explicitly state they aimed to avoid falling behind, as in "I gave resources to other players with lower scores to build up more, some to myself to try to keep up with the rest."

Additional analyses that investigate the relation between cooperativeness and outcomes are presented in S3 Text, S7 and S8 Figs.

## Discussion

In this study, we investigated how fair outcomes can emerge for heterogeneously endowed individuals. In order to delineate different concepts of fairness, we distinguished between dispersion (exaggerated rewards) and leveling (diminished distinctions) when it comes to the relationship between endowments and outcomes. We then conducted an online experiment to study how two simple structural conditions–the visibility of individual endowments and the visibility of individual outcomes–affect the fairness of outcomes in a network cooperation

game. We found that the visibility of outcomes produces equalization by decreasing the statistical dispersion of outcomes, while the visibility of endowments increases leveling, with benefits for the most disadvantaged.

By investigating emergent macro-level outcomes in social interaction groups, in contrast to unilateral allocation decisions, individual motivations, or subjective perceptions, our research builds upon, but also goes beyond, related work on fairness from the distributive justice, economic games, and social exchange literature. The interaction situations we model are marked by a tension between cooperation and self-interest, public information about endowments/ outcomes and lack of reputational information about behavior beyond direct interactions, ability to unilaterally change the outcome of any single individual but not to control the overall distribution of outcomes. Some of our findings were foreshadowed by prior work. As predicted by research on reciprocal exchange [74, 75], we observe relatively low levels of inequality in terms of the statistical dispersion of outcomes, even when the networks are endogenous. In alignment with findings on power use in negotiated exchange [4], we find that public knowledge of outcomes decreases inequality in reciprocal exchange interactions too. Remarkably, this occurs without the public accountability operationalized in distributive justice tasks [76, 77], since third parties could not observe giving decisions in our experiment.

Most novel and surprising are our findings on the effects of visible endowments. We observe similarly high statistical dispersion of outcomes when there is no information about others and when endowments are visible, but the reasons differ. Without any information, the statistical dispersion is high because the least endowed achieve corresponding outcomes and fall behind, in accordance with the principle of meritocracy. Whether this outcome is normatively desirable, of course, will depend on the nature of outcome-relevant characteristics under question. With visible endowments, outcomes are less predictable for everyone and although the least endowed have the opportunity to pull ahead, the most endowed do not lose their advantaged standing. The reason for this is that participants are more likely to give to the least endowed but also more likely to reciprocate gifts from better endowed individuals, as predicted by [58]. Thus, while participants give charitably to the less endowed in the short term, in the long term they seek to establish more durable relations with the better endowed. The better endowed gain, regardless of whether they choose to consistently defect or establish trustful relationships. From the other side, the worse endowed fail to capitalize on the promise of resource transfer from befriending the more resourceful because, ironically, long-term relations tend to skew distribution preferences towards equal rather than equitable division [29, 78].

Overall, we found that mutual help can diminish pre-existing and arbitrary differences among individuals. We also found that different conditions lead to different group outcomes in terms of fairness. For example, revealing only endowments and not outcomes is an acceptable solution according to Rawls' notion of fairness since the visible endowments will guarantee the lowest endowed a fairer deal and compensate for their "naturally occurring" disadvantage. On the other hand, if fairness is understood as equal outcomes regardless of endowments, then making outcomes visible will be the preferable scenario since it leads to outcomes with the lowest levels of statistical dispersion. If, in addition to equalized outcomes, the emphasis lies on equal opportunities, then making both outcomes and endowments visible offers the best solution. Finally, if the focus is on merit and rewards according to "inborn capabilities" or achieved characteristics, then the most preferable scenario is to keep information about endowments and outcomes private. This is the scenario that results in the clearest distinction between individuals with different endowments.

Although the theoretical framing of our study is broad and general, we acknowledg that our empirical investigation concerns a specific interaction situation: dyadic reciprocal exchange

with free partner selection, randomly assigned endowments, global information about others' endowments and/or outcomes, but lack of communication or observation of third-party behavior. Thus, our empirical results may not simply transfer to heterogeneity based on achieved characteristics, different types of social interactions, network dynamics, or information institutions. For instance, research using the N-person Prisoner's Dilemma game, constrained network updates, and local information finds opposing results: visible wealth entails higher Gini coefficients than the case of no information [19]. Similarly, our empirical results may not transfer to situations that allow for clique formation and coordinated group action. In our setup, when endowments were visible, multiple individuals independently helped the most disadvantaged to bring about equality of opportunity. However, we can imagine situations where interpersonal communication and social influence could result in stigmatizing and ostracizing the least endowed, making them poorer.

Another limitation of the experiment is related to the participant pool. Although our crowd-sourced American participants are relatively diverse, they still live in a WEIRD (Western, educated, industrialized, rich, and democratic) society and are thus more trustful towards strangers than the global population overall [79, 80]. Moreover, AMT workers are known to be particularly cooperative with each other due to in-group bias [81]. Consequently, our study likely overestimates the expected level of cooperation and prominence of fairness considerations compared to less cohesive and non-Western populations. Nevertheless, even if the strength of effects may not extend beyond low-competition settings with good levels of group solidarity and group identity, the direction of effects we hypothesize and find likely holds. Additional research could help determine the scope conditions for our empirical results.

As a further limitation, since we specifically designed the research to test hypotheses at the group level, our empirical study is not best suited to elicit the motivation and intention behind participants' decisions and behavior. We employed observational study methods and coded participant survey responses to identify behavioral proclivities, but we can only interpret these as charity, reciprocity, and inequity aversion. Since the research was not designed to provide direct and causal evidence for individual behavior and motivations, our conclusions regarding these aspects remain tentative. Future research should employ customized experimental designs to investigate how perceptions of fairness and behavioral motivations change when inequality is more or less salient. The work of Molina et al. [82] and Sands [83, 84] exemplify how this can be accomplished with two-player games and field experiments.

Nevertheless, the individual-level analyses spotlighted an important aspect of our own experimental design: the global information and unconstrained partner selection in the experiment allowed our participants to act globally. According to their survey responses, some participants did not just consider their own and their partners' outcomes, but acted towards an overall group outcome: ". . . try to make sure that no other little place got left too far behind"; I shared my resources with those who seemed to need it most"'; "I tried to spread the wealth around." This behavior would be impossible if information was constrained to the endowments and/or outcomes of one's interaction partners only. Thus, future research should establish the extent to which our findings depend on the assumption of global information instead of local. Specifically, local information may weaken the effects from the visibility of outcomes and endowments or, if combined with homophily, potentially even reverse them.

Another promising direction for further research is to allow players to invest in increasing their endowment, similarly to how people invest in human capital via education and training. This idea is the reason why we refrained from restricting our definition of endowments to ascribed characteristics only. As we discussed, outcome-relevant characteristics such as cognitive ability are to some extent ascribed in terms of genetic inheritance or advantageous upbringing but can also be cultivated. Decisions to invest in one's endowment will trigger

cumulative advantage processes, which make individual outcomes more sensitive to chance and early investment decisions. We know that cumulative advantage processes tend to make the rich richer and the poor poorer and thus increase the dispersion of outcomes [39]. However, we are yet to find out how they affect social interactions and ultimately, leveling. The experiment by Gächter et al. [85] paves the way for research in this direction.

Despite the limitations and need for further research, our study offers valuable insights. First and foremost, our broader conceptualization of macro-patterns of fairness is agnostic to research design and methods and can be utilized for a variety of problems and approaches, including eliciting individual allocation and redistribution preferences and qualifying inequality in organizations or the general population. Second, our gamification approach to designing and conducting experiments is a promising way to engage human participants online and we hope experimental social scientists will pick it up to develop and exploit further. Finally, although with certain caveats, we can draw parallels between the simplified and abstract system of exchange we study experimentally and face-to-face and online non-competitive communities where members exchange personal advice, direct help, rival knowledge, and one-on-one training that improve productivity and performance. For instance, our findings suggest that sharing students' grades in schools and sharing employees' productivity statistics in companies could redirect members' attention to those who are falling behind and result in more equalized performance and outcomes. Similarly, opening up about hidden disabilities might increase the help rendered to the most disadvantaged and provide them with better opportunities to improve their lot. These effects, however, may not take place and even flip if communication, social influence, and intergroup dynamics lead to the stigmatization and ostracism of the disadvantaged. We need more research to establish the precise structural conditions and delimit the specific social contexts where differences do not automatically imply inequality.

## Supporting information

**S1 Table. Results from discrete-time survival analysis for the likelihood to drop out from the game.** The analysis is implemented as a multi-level logistic regression model with random intercepts by group and participant.
(PDF)

**S1 Fig.** Screenshots from the interactive tutorial (top) and the 19th round of a 17-player game in the treatment with visible endowments (bottom). The current player has an endowment of two but has not allocated their resources to anyone yet. The orange arcs indicate the player's interactions from the previous round with animations showing the number of resources given or received. The block in front is highlighted because the current player's cursor is pointing to it. If the player clicks on the block, one of their two resources will transfer to this block.
(PDF)

**S2 Fig. The least endowed earn significantly more when endowments and/or outcomes are visible (E, O, and EO) compared to when no information about others is available (NI).** The figure shows boxplots and datapoints of players' final score depending on their endowment for each group in each treatment. The shaded areas show what score a player of the specified endowment will get if they either do not cooperate at all or invest all their resources in partners who reciprocate perfectly. This is the situation with $c_i = c_{-i} = 0$ and $c_i = c_{-i} = r_i$, where $c_i$ is the number of resources player $i$ gives to others, $c_{-i}$ is the number of resources other players give to $i$, and $r_i$ is the sum of resources player $i$ has access to over the game.
(PDF)

**S3 Fig. The least endowed cooperate significantly more than the others when endowments are visible (O and EO in top panel) and earn significantly more when endowments and/or outcomes are visible (E, O, and EO in bottom panel) compared to when no information about others is available (NI).** The figures show boxplots of players' cooperativeness (top) or final score (bottom) for players with different endowments. Cooperativeness is defined as the proportion of available resources invested in others. The asterisk brackets show statistically significant pairwise differences tested in individual-level linear regression models with random intercepts by group: $^*$ $p<0.05$,$^{**}$ $p<0.01$,$^{***}$ $p<0.001$.
(PDF)

**S4 Fig. Reciprocity is higher when no information is available (NI) especially compared to the treatments with visible outcomes (O and EO).** The figure shows the network reciprocity for the groups in the four treatments. Reciprocity is estimated as the ratio of the number of edges pointing in both directions to the total number of edges in the network. The shaded areas correspond to the 95% confidence intervals around the means.
(PDF)

**S5 Fig.** When endowments are visible (E), the least endowed prefer to give to the most endowed in the first round (top) but this charitable giving by endowment does not persist throughout the game (bottom). The heatmaps show the difference between the proportion of resources given in the first round (top) or overall in the game (bottom) by players with the giver's endowment to players with the recipient's endowment and the proportion expected to be given if the selected number of resources were allocated at random.
(PDF)

**S6 Fig. When outcomes are visible (O and EO), all players tend to give more to the poor.** The heatmap shows the difference between the proportion of resources given by players with the giver's score to players with the recipient's score and the proportion expected to be given if the resources were allocated at random. The givers' and recipients' scores are standardized per game round and the cells show the mean over the 200 game rounds (20 rounds × 10 games per treatment). The highest score values include players with score that is two standard deviations or higher than the mean. The extreme values in the leftmost column and the bottom row are due to the small number of observations of players with score that is two standard deviations below the mean.
(PDF)

**S7 Fig. The outcomes are more dispersed in the treatment without information (NI) because the least endowed (Endowment = 2) remain poor, and in the treatment with visible endowments (E), because anyone, regardless of their endowment level, can end up rich or poor.** The less dispersed outcomes in the treatments with visible outcomes (O and EO) come at the expense of well-endowed (Endowment = 6) altruistic players. The figure plots players' scores at the end of the game against their cooperativeness $c_i$, measured as the proportion of resources $r_i$ that the player chooses to invest in others. The curves are the best fitting quadratic polynomials for each endowment level. The shaded areas show the expected score for a player who does not cooperate ($c_i = 0$) and receives nothing from others ($c_{-i} = 0$, bottom left) or receives the equivalent of half of their own resources ($c_{-i} = r_i/2$, top left), and for a player who invests all their own resources in others ($c_i = r_i$) and receives exactly the same back ($c_{-i} = r_i$, top right) or just half of that ($c_{-i} = r_i/2$, bottom right).
(PDF)

**S8 Fig. The outcomes are most dispersed and thus less predictable for a particular endowment level when endowments are visible but not outcomes (E).** The figure shows the Gini coefficient of outcomes for individuals with the particular endowment by group. The asterisk brackets show statistically significant pairwise differences tested with the Mann-Whitney $U$ test: * $p<0.05$,** $p<0.01$,*** $p<0.001$.
(PDF)

**S1 Text. Game design.**
(PDF)

**S2 Text. Temporal exponential random graph models.**
(PDF)

**S3 Text. Supplementary analyses.**
(PDF)

## Acknowledgments

The authors would like to acknowledg the Science at Home team for their constructive design input for the game. We are also grateful to the two anonymous reviewers for their most constructive feedback and suggestions.

## Author Contributions

**Conceptualization:** Milena Tsvetkova, Oana Vuculescu, Claudia Wagner.

**Data curation:** Milena Tsvetkova, Oana Vuculescu.

**Formal analysis:** Milena Tsvetkova, Petar Dinev.

**Funding acquisition:** Milena Tsvetkova, Jacob Sherson, Claudia Wagner.

**Investigation:** Milena Tsvetkova.

**Methodology:** Milena Tsvetkova, Jacob Sherson, Claudia Wagner.

**Project administration:** Milena Tsvetkova, Oana Vuculescu, Claudia Wagner.

**Resources:** Jacob Sherson.

**Software:** Jacob Sherson.

**Visualization:** Milena Tsvetkova.

**Writing – original draft:** Milena Tsvetkova.

**Writing – review & editing:** Milena Tsvetkova, Oana Vuculescu, Petar Dinev, Jacob Sherson, Claudia Wagner.

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
