## [Decision Letter · Decision Letter 0]

24 May 2022

PONE-D-22-08604Inequality and fairness with heterogeneous endowmentsPLOS ONE

Dear Dr. Tsvetkova,

Thank you for submitting your manuscript to PLOS ONE. After careful consideration, we feel that it has merit but does not fully meet PLOS ONE’s publication criteria as it currently stands. Therefore, we invite you to submit a revised version of the manuscript that addresses the points raised during the review process.

We look forward to receiving your revised manuscript.

Kind regards,

Alberto Antonioni, PhD

Academic Editor

PLOS ONE

Journal Requirements:

“This research was made possible through the generous support of the Volkswagen Foundation (Grant Ref. 92 173). The game development was partially supported by the Carlsberg Foundation (Grant no. CF16–0593). The funders had no role in the study design, data collection and analysis, decision to publish, or preparation of the manuscript. We would also like to acknowledge the Science at Home team for their constructive design input”

“This research was made possible through the generous support of the Volkswagen Foundation (Grant Ref. 92 173). The game development was partially supported by the Carlsberg Foundation (Grant no. CF16–0593). The funders had no role in the study design, data collection and analysis, decision to publish, or preparation of the manuscript.”

Additional Editor Comments:

Considering the numerous constructive comments given by both reviewers, I invite the authors to address them improving the presentation of their study.

Reviewers' comments:

Reviewer's Responses to Questions

**Comments to the Author**

1. Is the manuscript technically sound, and do the data support the conclusions?

Reviewer #1: Yes

Reviewer #2: Yes

2. Has the statistical analysis been performed appropriately and rigorously? 

Reviewer #1: Yes

Reviewer #2: I Don't Know

3. Have the authors made all data underlying the findings in their manuscript fully available?

Reviewer #1: Yes

Reviewer #2: No

4. Is the manuscript presented in an intelligible fashion and written in standard English?

Reviewer #1: Yes

Reviewer #2: Yes

5. Review Comments to the Author

Reviewer #1: This is a most interesting paper that synthesizes ideas from political theory, sociology, social psychology, and behavioral economics on fairness perceptions, sharing behavior, and related group-level outcomes of material inequality. It tests predictions using rich experimental data both at the group level (which I found most interesting) and at the individual level (using a sophisticated ERGM methodology). Overall, the design and data are well used and get at some long standing questions in novel ways. I have a number of questions and comments, which I hope are useful for the authors. I order my comments following their text:

Abstract: It was not clear to me how the first half of the Abstract relates to the research questions and the manuscript’s main thrust about visibility of endowments and outcomes. Also it remained unclear how the functional relationship between endowments and outcomes is investigated. Isn’t this more of a classification exercise than an “investigation”? The main idea of the experiment also remained nebulous, isn’t the “network cooperation experiment” about partner selection and sharing? Please clarify these points in the Abstract.

Intro: The authors state that their work “contributes in several ways” (p. 3), but precisely what really is new, what new insights are yielded, and why this research is important has, in the current form, not been made crystal clear. Statements at the start of the Related Work section and in the Discussion section made it clearer, and I suggest moving these sentences into the Intro to carve out the contribution. Small point: On p. 3 the authors speak of combining “political theory, social psychology, and behavioral economics” and then they cite several sociology papers in the same paragraph.

Theory section: The authors use a wide definition of what they call “endowments,” including privileged social origin and luck (p. 5, 6). Then they posit that meritocracy is characterized by a linear mapping of endowments to outcomes. This contradicts the typical understanding of meritocracy as a state of the social world where rewards are determined by performance (merit as opposed to privilege and luck).

The authors may want to explain in a bit more detail the seminal work by Lynn, Podolny and Tao (2009) which they use to motivate their theorizing on dispersion and leveling; most importantly perhaps: what are the conditions (e.g. uncertainty) under which dispersion or leveling occurs. To what degree is this study an experimental test of Lynn et al.’s theoretical work? A drawback for the usefulness of Lynn et al.’s framework here is that the authors’ experimental design does not allow for increasing-return processes (p. 20) that in my reading is crucial in Lynn et al.’s model to disconnect outputs (e.g. status) from inputs (e.g. quality).

To be able to better follow the derivation of hypotheses, and the fact that the asserted micro-level mechanisms pertain to both the choice of interaction partners and the choice about giving that together bring about group outcomes, it would be helpful to learn about the experimental situation briefly already in the Intro. The descriptions of human behavior that lead up to each hypothesis sound overly deterministic while at times it is unclear whether hypotheses follow strictly from theoretical claims or from prior empirical results.

Materials and Methods: The gamified experimental setting has not been motivated in the front end of the paper. Is this something of extra methodological interest (e.g. increasing external validity), a specific contribution, or is it just about subject engagement? How does it affect comparability to prior lab research? Cliff’s delta is a non-standard measure, at least in large parts of the social sciences. What are its benefits over more common (non-parametric) measures and are the results on leveling robust to changing the measure?

Results: I value highly the focus of group-level outcomes over the determinants of individual sharing behavior. This is where the manuscript makes its strongest contribution. I wondered why there is no result plot paralleling Fig. 1 (or a linear correlation analysis) despite statements like “This indicates that, in the absence of information, a meritocratic system emerges in which endowments and outcomes are linearly related” (p. 21). In the context of the micro-level ERGM analysis more interpretations with regard to partner selection would be interesting. The observational (or qualitative) study surrounding Fig. 5 nicely illustrates and adds to what has been shown in the main results.

Reviewer #2: This manuscript introduces an experimental design to understand how individual interactions can produce patterns of group-level outcomes that refer to traditional fairness concepts, when the visibility of endowments and partial outcomes is manipulated. These group-level outcomes emerge as a function of complex individual interactions that are not necessarily linked to the group and, in this sense, they are emergent. The manuscript was very interesting and it presented some important findings. However, I also think that it needs more work and some clarifications for the manuscript to reach its full potential. I hope the following comments help the authors improve their work.

The authors describe several group-level outcomes in terms of dispersion and leveling, the combination of which produces traditional fairness concepts at the group leve, such as “equality of opportunity” or “wealth concentration.” I found this discussion of how dispersion and leveling produce different fairness systems very interesting and one of the main contributions of the manuscript. Nonetheless, the connection of dispersion and leveling to what Lynn and colleagues (2009) argue in their article is somewhat confusing. The reason is that Lynn et al (2009) discuss those mechanisms to explain how social status can deviate from underlying heterogeneity in quality (e.g., distribution of skills). It is unclear to me how social status connects to the dispersion or leveling of outcomes in this theoretical discussion, especially considering that the mechanisms discussed by Lynn et al (2009) are mostly driven by social influence and, in my understanding, there is no social influence manipulation in the design. How is social influence addressed in the experimental design and how it relates to the concepts discussed by Lynn et al? Alternatively, I also thought that the authors were not really interested in social status per se but they were inspired by Lynn et al (2009) to discuss different principles of fairness as if there were dispersion and leveling in outcomes, borrowing these concepts from Lynn et al. If this is the case, the authors need to make this connection clearer and perhaps place the reference to Lynn et al on a footnote as status dynamics are not directly relevant for the manuscript’s discussion about fairness.

Moreover, the manuscript makes the claim that when there is no information about outcomes and endowments, a meritocratic distribution of final outcomes takes place: those who have more get more and those who have less get less. This point is unclear because it really depends on the nature of the endowments and the authors say that they do not want to specify whether endowments correspond to an ascribed or achieved characteristic (p. 6) – although they also state in the conclusion that they restrict “the definition of endowments to ascribed characteristics only” (p. 30), which makes the statement about meritocracy even more puzzling. In my understanding of the findings, however, I think that only if endowments are understood as achieved, one can see how the no-information condition can produce a meritocratic system. In other situations, we would need to accept some relationships between endowments and outcomes as meritocratic when they are clearly non-meritocratic – e.g., if we assume that endowments are similar to inheritances. I think the argument about meritocracy needs some clarification at different parts on the manuscript, particularly when the nature of endowments is discussed.

In the conclusion, the authors suggest some policy implications when they say that outcomes should be visible in some social settings such as the workplace and the school. I presume this implication is because one of the findings is that the visibility of outcomes decreases dispersion (as shown in figure 2) and because this pattern is driven mostly by “inequity aversion” and “helping the poor” (as shown in figure 4). However, it is unclear from the findings how other social mechanisms may interact with the visibility of outcomes in the social settings for which these policy designs are suggested. For instance, one could imagine a situation where groups emerge based on performance and low-performers are ostracized by others, not integrated as suggested by the findings. This restriction (i.e., the absence of coalition formation) may be a scope condition for a finding that was produced with no intergroup dynamics. This point deserves some discussion in connection with the proposed implications.

In a similar vein, it was interesting to see that no hypothesis referred to the dispersion of outcomes directly, even though dispersion is probably the most similar scenario to current distributions of income and wealth in the world: for developed countries, there is extreme wealth concentration and for developing countries, there is extreme inequality, following the schema in figure 1. The authors preempt important differences with other studies that produce wealth concentration in the “Related work” section in the front end and then go back to these differences in the conclusion. I believe that it is also important to say something about this when the hypotheses are discussed because the absence of hypotheses that relate to dispersion is really an unsaid statement about these findings not being expected, given the mechanisms of reciprocity, inequity aversion, and charity. Also, the conclusion asserts that “our empirical results may not simply transfer to different types of social interactions, network dynamics, or information institutions” and the research by Nishi et al (2015) that relates to wealth concentration is then discussed as an instance of this limitation. I think that the conclusion would be stronger if the authors mention instead what types of interactions their findings are informative about, and perhaps only then make a reference to Nishi et al (2015) as part of the scope conditions of their design (rather than as a limitation).

A few other comments are the following. First, the main analysis in figures 2 and 3 are based on 10 data points for each experimental condition. I wonder if the authors can say something about this low-N problem, why we should not be concerned that the statistically significant differences are not due to noise (plotting confidence intervals can be better than presenting p-values), and why they decided to put aside the temporal evolution of the groups in the main analysis (I see that figure C1 includes the temporal evolution of the games). This could have increased the sample size and open up the possibility to use other modeling strategies to account for the more complex correlational structure between the variables of interest. Second, it seems to me that endowments can be inferred in the O condition, similarly to what happens in real life where we may have clear information about wealth or income and no information about endowments but try to infer what the distribution of endowments is. This could explain why this condition is so similar to the EO condition in several of the findings presented on the manuscript, including individual behaviors. Is this inference possible? Third,

Minor comments

• I see that the authors talk about “the dispersion of the outcome” in the appendix to refer to the variance of the outcome (e.g., “the outcomes are more/most dispersed…” in figures C4 and C5 or “the dispersion of the outcomes is driven by the low scores…” in p. 8, Appendix C). I found this language confusing as they introduce the theoretical concept of “dispersion” in regard to fairness and inequality in the front end.

• In page 24, the authors say that they manually coded answers to the question about individual strategies. Can they say a bit more about how this task was actually performed? For instance, is there intercoder reliability?

6. PLOS authors have the option to publish the peer review history of their article (what does this mean?). If published, this will include your full peer review and any attached files.

Reviewer #1: No

Reviewer #2: No

---

## [Author Response · Author response to Decision Letter 0]

7 Jul 2022

Please refer to attached file with detailed response.

---

## [Decision Letter · Decision Letter 1]

19 Sep 2022

PONE-D-22-08604R1Inequality and fairness with heterogeneous endowmentsPLOS ONE

Dear Dr. Tsvetkova,

Thank you for submitting your manuscript to PLOS ONE. After careful consideration, we feel that it has merit but does not fully meet PLOS ONE’s publication criteria as it currently stands. Therefore, we invite you to submit a revised version of the manuscript that addresses the points raised during the review process.

We look forward to receiving your revised manuscript.

Kind regards,

Alberto Antonioni, PhD

Academic Editor

PLOS ONE

Journal Requirements:

Additional Editor Comments:

Considering both reviewers' positive evaluation, the manuscript can be now considered for publication after a minor revision to address Reviewer 2's last remarks. In particular, there are some parts that can still improve the presentation of the study. In case the author satisfactorily addresses these comments an additional round of revision won't be needed before proceeding for publication.

Reviewers' comments:

Reviewer's Responses to Questions

**Comments to the Author**

1. If the authors have adequately addressed your comments raised in a previous round of review and you feel that this manuscript is now acceptable for publication, you may indicate that here to bypass the “Comments to the Author” section, enter your conflict of interest statement in the “Confidential to Editor” section, and submit your "Accept" recommendation.

Reviewer #1: All comments have been addressed

Reviewer #2: (No Response)

2. Is the manuscript technically sound, and do the data support the conclusions?

Reviewer #1: Yes

Reviewer #2: Yes

3. Has the statistical analysis been performed appropriately and rigorously? 

Reviewer #1: Yes

Reviewer #2: Yes

4. Have the authors made all data underlying the findings in their manuscript fully available?

Reviewer #1: Yes

Reviewer #2: Yes

5. Is the manuscript presented in an intelligible fashion and written in standard English?

Reviewer #1: Yes

Reviewer #2: Yes

6. Review Comments to the Author

Reviewer #1: I applaud the authors for their thorough revision that addressed all my comments. I feel the manuscript has greatly improved and I now see it fit for publication.

Reviewer #2: I appreciate all the work that went into preparing this revision, and I commend the authors for this version. Most of my previous concerns were addressed and clarified. I also like how the introduction reads now: it states clearly what the question is and what the contributions are. However, a few concerns remain. I leave my comments to the authors below, which I hope are helpful to the authors.

• The text is much clearer about the difference between dispersion and leveling. However, the new version makes me think that the reference to Lynn et al. (2009) is unnecessary. If anything, it adds more confusion to the authors’ explanation. While the concept of dispersion used in the manuscript is similar to the use in Lynn et al., the concept of leveling does not entirely match the use of ‘rank reordering’ in Lynn et al. I think the slopes in figure 1 should be negative to reflect the concept of rank reordering more appropriately. Since this comparison is not crucial for the manuscript’s argument, I would suggest that the authors either eliminate that paragraph or leave it in a footnote for readers curious about the parallels.

• The authors define poverty and wealth in terms of dispersion in the front end, not in terms of Gini. Consequently, the authors should not conclude that the treatments do not produce poverty or wealth concentration because the Gini coefficients are low. They say this happens because their interaction situation model leads to lower Gini coefficients by design (p. 25). It would be better to state that the treatments do not produce poverty or wealth concentration because no condition leads to steeper slopes (as shown in fig. 2).

• I still have questions about the concept of meritocracy. I understand what the authors are trying to do, and I think that stating explicitly how they are using meritocracy in this setting helps the reader offset some of the implicit notions they have about the notion of meritocracy when reading the manuscript. The critical problem is that some deviations from the meritocratic scenario imply different uses of the concept of ‘endowment.’ For instance, when the authors describe equality of opportunity, they say it means that “individuals should not be limited by characteristics ascribed at birth.” Hence, even if the reader is willing to concede that endowments can refer to ascribed or achieved characteristics, we are forced to think about endowments as ascribed—not achieved— when we compare the meritocratic condition with equality of opportunity. In the E condition, the Cliff’s delta statistic remains low (in fig. 4) because participants likely interpret endowments as ascribed (given that they are exogenously assigned). Why would participants give charitably to the least endowed (as stated in the conclusion; p. 33) if they do not perceive endowments as ascribed? In short, although it is clear conceptually why endowments could be regarded as ascribed or achieved, the experimental design forces participants (and readers) to understand endowments only as ascribed.

• In figure 2, the authors state that “when outcomes are visible, they end up less equalized and less variable” (p. 23). I only see this for participants with 2 and 4 endowments, however. The score difference for the most endowed is highly variable, especially in the O condition. Am I missing something here?

• Moreover, I found it very interesting to compare the results in terms of dispersion and leveling (figs. 3 and 4) with the distribution of scores across endowments in fig. 2. I believe that crossing the information from these figures renders a couple of insights that are not highlighted enough in the manuscript. For instance, in the conclusion, the authors say that the visibility of outcomes leads to equality due to lower Gini (p. 32 and p. 33). But this understates the shape of final scores in figure 2 and the variability of Cliff’s delta between conditions O and EO. Indeed, fig. 2 suggests that both conditions resemble equality of opportunity. Fig. 4 shows that this form of equality in condition EO occurs because the lower class (with two endowments) becomes similar to the middle class (with four endowments) regarding how opportunities map to outcomes. In condition O, equality of opportunity is also achieved, but endowments seem more consequential for the lower class (i.e., there is less mobility), as observed in fig. 4. This comparison makes me think that the visibility of outcomes and the visibility of endowments are not as different as implied in the conclusion (e.g., p. 33.). More generally, the addition of fig. 2 makes the reader wonder whether the findings for group-level fairness can be described in terms of the fairness concepts represented in fig. 1.

7. PLOS authors have the option to publish the peer review history of their article (what does this mean?). If published, this will include your full peer review and any attached files.

Reviewer #1: No

Reviewer #2: No

---

## [Author Response · Author response to Decision Letter 1]

12 Oct 2022

Please see the attached file entitled Response to Reviewers.

---

## [Editor Report · Decision Letter 2]

17 Oct 2022

Inequality and fairness with heterogeneous endowments

PONE-D-22-08604R2

Dear Dr. Tsvetkova,

We’re pleased to inform you that your manuscript has been judged scientifically suitable for publication and will be formally accepted for publication once it meets all outstanding technical requirements.

Kind regards,

Alberto Antonioni, PhD

Academic Editor

PLOS ONE

Additional Editor Comments (optional):

The manuscript can be now accepted for publication on PLoS ONE without further reviews.
---

## [Editor Report · Acceptance letter]

21 Oct 2022

PONE-D-22-08604R2 

Inequality and fairness with heterogeneous endowments 

Dear Dr. Tsvetkova:

I'm pleased to inform you that your manuscript has been deemed suitable for publication in PLOS ONE. Congratulations! Your manuscript is now with our production department. 

Kind regards, 

on behalf of

Dr. Alberto Antonioni 

Academic Editor

PLOS ONE